# Mitigating Barren Plateaus in Quantum Neural Networks via an AI-Driven Submartingale-Based Framework

## Abstract

In the era of noisy intermediate-scale quantum (NISQ) computing, Quantum Neural Networks (QNNs) have emerged as a promising approach for various applications, yet their training is often hindered by barren plateaus (BPs), where gradient variance vanishes exponentially in terms of the qubit size. Most existing initialization-based mitigation strategies rely heavily on pre-designed static parameter distributions, thereby lacking adaptability to diverse model sizes or data conditions. To address these limitations, we propose AdaInit, a foundational framework that leverages generative models with the submartingale property to iteratively synthesize initial parameters for QNNs that yield non-negligible gradient variance, thereby mitigating BPs. Unlike conventional one-shot initialization methods, AdaInit adaptively explores the parameter space by incorporating dataset characteristics and gradient feedback, with theoretical guarantees of convergence to finding a set of effective initial parameters for QNNs. We provide rigorous theoretical analyses of the submartingale-based process and empirically validate that AdaInit consistently outperforms existing initialization methods in maintaining higher gradient variance across various QNN scales. We believe this work may initiate a new avenue to mitigate BPs.

## 1 Introduction

In recent years, there have been significant advancements in quantum computing, particularly with the advent of noisy intermediate-scale quantum (NISQ) devices (Preskill, 2018). Within this research landscape, quantum neural networks (QNNs), which integrate quantum circuits with classical deep-learning layers, have been widely applied in various domains, such as quantum machine learning (Stein et al., 2021), quantum chemistry and materials modeling (Kandala et al., 2017), and combinatorial optimization (Farhi et al., 2014). However, recent studies (Ortiz Marrero et al., 2021; Cerezo et al., 2021)

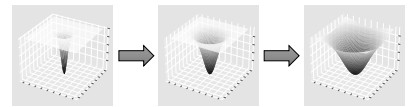

Figure 1: Example of BPs' mitigation process. A plateau-dominated loss landscape ($1^{st}$ image), a.k.a. BPs, could be gradually recovered to the normal case ($3^{rd}$ image) after mitigation.

reveal that the performance of QNNs may be hindered due to gradient issues, such as barren plateaus (BPs), a kind of gradient issue that the initialization of QNNs might be trapped on a flattened landscape at the beginning of training. McClean et al. (2018) first systematically investigate BPs and affirm that the gradient variance will exponentially decrease when the QNNs satisfy the assumption of the 2-design Haar distribution. Under these circumstances, most gradient-based training approaches would fail. To better illustrate the BPs' mitigation process, we present an example in Fig. 1.

Numerous studies have been devoted to mitigating the BPs, among which initialization-based strategies have proven particularly effective by using well-designed distributions to initialize QNNs' parameters (Sack et al., 2022). However, most existing initialization-based strategies, such as GaInit (Zhang et al., 2022) or BeInit (Kulshrestha & Safro, 2022), rely on a one-shot generation of static initial parameters, which suffer from two critical limitations: (i) they often depend on idealized distribution assumptions, and (ii) they lack adaptability when scaling to various model sizes or data conditions. These limitations are particularly problematic where BPs hinder effective QNNs' training.

To address the limitations, we propose **AdaInit**, an iterative framework that combines generative modeling with the submartingale property. We prove its convergence, demonstrating that AdaInit **Ada**ptively generates effective **Init**ial parameters for QNNs. These parameters ensure non-negligible gradient variance at the beginning of training, thereby mitigating BPs. Beyond the theoretical guarantee, AdaInit intuitively can help QNNs locate initial parameters in the "non-flat" loss landscape, which prevents training from being trapped from the start. Our method is grounded by **two key ideas**. First, instead of statically pre-designing the initialization distribution, we leverage generative models, such as large language models (LLMs), to synthesize candidate parameters based on dataset descriptions and prior gradient feedback. This allows the initializer to actively explore the parameter space and adaptively refine the candidate. Second, by modeling the iterative process as a submartingale, we provide a theoretical guarantee that this process will almost surely converge within a finite number of iterations to initial parameters that yield non-negligible gradient variances. We model such a process because the submartingale property can depict the trend of expected improvement in gradient variance and guarantee convergence in finite time. Besides theoretical analysis, we conduct extensive experiments to validate the effectiveness of our proposed framework across various model scales. The results reveal that our framework can maintain higher gradient variances against three classic initialization methods and two popular initialization-based strategies for mitigating BPs. Overall, our primary contributions can be summarized as:

- We propose a new artificial intelligence (AI)-driven submartingale-based framework, AdaInit, for mitigating BPs. To the best of our knowledge, we open a new avenue to leverage LLMs with submartingale property to model QNNs' initial parameters for mitigating BPs.
- We theoretically analyze the submartingale property of the iterative process and rigorously prove its supremum and expected hitting time.
- Extensive experiments across various model scales demonstrate that as the model size of QNNs increases, our framework can maintain higher gradient variances against classic initialization methods.

## 2 RELATED WORK

McClean et al. (2018) first investigated BP phenomena and demonstrated that under the assumption of the 2-design Haar distribution, gradient variance in QNNs will exponentially decrease to zero at the beginning of training as the model size increases. In recent years, enormous studies have been devoted to mitigating BP issues in QNNs (Qi et al., 2023). Cunningham & Zhuang (2024) categorize most existing studies into the following five groups. (i) Initialization-based strategies initialize model parameters with various well-designed distributions in the initialization stage (Grant et al., 2019; Sack et al., 2022; Mele et al., 2022; Grimsley et al., 2023; Liu et al., 2023; Park & Killoran, 2024; Kashif et al., 2024; Shang & Shi, 2025). (ii) Optimization-based strategies address BP issues and further enhance trainability during optimization (Ostaszewski et al., 2021; Suzuki et al., 2021; Heyraud et al., 2023; Liu et al., 2024; Sannia et al., 2024). (iii) Model-based strategies attempt to mitigate BPs by proposing new model architectures (Li et al., 2021; Bharti & Haug, 2021; Du et al., 2022; Selvarajan et al., 2023; Tüysüz et al., 2023; Kashif & Al-Kuwari, 2024). (iv) To address both BPs and saddle points, Zhuang et al. (2024) regularize QNNs' model parameters via Bayesian approaches. (v) Rappaport et al. (2023) measure the BP phenomenon via various informative metrics.

## 3 PRELIMINARIES

In this section, we first introduce the preliminary background about the basics of quantum computing and barren plateaus, and then present the necessary tools from probability theory.

**Quantum Basics.** A quantum state can be seen as a unit vector $|\psi\rangle$ in a complex Hilbert space $\mathcal{H}^m$, satisfying the normalization condition $\langle\psi|\psi\rangle = 1$. We use the Dirac bra-ket notation, where ket $|\psi\rangle$, bra $\langle\psi|$ denote a column vector in $\mathbb{C}^m$ and its Hermitian conjugate (conjugate transpose), respectively. Any $|\psi\rangle$ can be written as a linear combination of computational basis states, $|\psi\rangle = \sum_{i=1}^{m} c_i|i\rangle$, where $c_i \in \mathbb{C}$ are called the *amplitudes* of the basis states $|i\rangle$. Given two states $|\psi\rangle \in \mathcal{H}^m$ and $|\phi\rangle \in \mathcal{H}^n$, their inner product can be denoted by $\langle\psi|\phi\rangle \triangleq \sum_i \psi_i^{\dagger}\phi_i$, whereas their tensor product can be denoted by $|\psi\rangle \otimes |\phi\rangle \in \mathcal{H}^{m \times n}$. If we measure state $|\psi\rangle = \sum_{i=1}^{m} c_i|i\rangle$ on a computational basis, we will obtain $i$ with probability $|c_i|^2$ and the state will collapse into $|i\rangle$ after measurement.

**Variational Quantum Circuits (VQCs)**, whose model architecture is constructed solely from parameterized quantum circuits without interleaving classical neural network layers, play a core role in quantum neural networks (QNNs) (Mitarai et al., 2018; Mari et al., 2020). Typical VQCs consist of a finite sequence of unitary gates $U(\boldsymbol{\theta})$ parameterized by $\boldsymbol{\theta} \in \mathbb{R}^{LNR}$, where $L$, $N$, and $R$ denote the number of layers, qubits, and rotation gates. $U(\boldsymbol{\theta})$ can be formulated as:

$$U(\boldsymbol{\theta}) = U(\theta_1, ..., \theta_L) = \prod_{l=1}^{L} U_l(\theta_l), \tag{1}$$

where $U_l(\theta_l) = e^{-i\theta_l V_l}$, $V_l$ is a Hermitian operator.

QNNs, which are built by wrapping neural network layers with VQCs, can be optimized using gradient-based methods. To optimize QNNs, we first define the loss function $E(\boldsymbol{\theta})$ of $U(\boldsymbol{\theta})$ as the expectation over Hermitian operator $H$:

$$E(\boldsymbol{\theta}) = \langle 0|U(\boldsymbol{\theta})^{\dagger} H U(\boldsymbol{\theta})|0\rangle. \tag{2}$$

Given the loss function $E(\boldsymbol{\theta})$, we can further compute its gradient by the following formula:

$$\partial_k E \equiv \frac{\partial E(\boldsymbol{\theta})}{\partial \theta_k} = i\langle 0|U_{-}^{\dagger} \left[V_k, U_{+}^{\dagger} H U_{+}\right] U_{-}|0\rangle, \tag{3}$$

where we denote $U_{-} \equiv \prod_{l=0}^{k-1} U_l(\theta_l)$ and $U_{+} \equiv \prod_{l=k}^{L} U_l(\theta_l)$. Also, $U(\boldsymbol{\theta})$ is sufficiently random s.t. both $U_{-}$ and $U_{+}$ (or either one) are independent and match the Haar distribution up to the second moment.

**Barren Plateaus (BPs)** are first investigated by McClean et al. (2018), who demonstrate that the gradient variance $\text{Var}[\partial E]$ of QNNs at the beginning of training will exponentially decrease as the number of qubits $N$ increases when the random QNNs match 2-design Haar distribution. This exponential pattern can be approximated as:

$$\text{Var}[\partial E] \propto 2^{-2N}. \tag{4}$$

The Eq. (4) indicates that $\text{Var}[\partial E]$ will approximate zero when the number of qubits $N$ is very large, i.e., most gradient-based approaches will fail to train QNNs in this case.

Based on the above description, we formally state the problem that we aim to solve as follows:

**Problem 1.** *By leveraging a generative AI (GenAI) model, such as an LLM, we refine posterior with adaptive prompting, i.e., iteratively generate effective initial parameters $\boldsymbol{\theta}_0^*$ for a QNN that yields non-negligible gradient variance $\text{Var}[\partial E]$, thereby mitigating barren plateaus (BPs).*

**Tools from Probability Theory.** Below, we review the definition of martingale (submartingale), along with key tools relevant to our work. We adapt the descriptions from (Williams, 1991; Freeman & Stephenson, 2025).

**Definition 1** (Martingale, (Williams, 1991)). *Let $\{M^{(t)}\}_{t \geq 1}$ be a stochastic process w.r.t. a filtration $\{\mathcal{F}^{(t)}\}_{t \geq 1}$ on a probability space $(\Omega, \mathcal{F}, P)$. The process $\{M^{(t)}\}$ is called a martingale if (i) $\{M^{(t)}\}$ is adapted, (ii) $\mathbb{E}[|M^{(t)}|] < \infty$, for $\forall t \in \mathbb{Z}^+$, (iii) $\mathbb{E}[M^{(t+1)} \mid \mathcal{F}^{(t)}] = M^{(t)}$, almost surely for $\forall t \in \mathbb{Z}^+$.*

*If (iii) is replace by $\mathbb{E}[M^{(t+1)} \mid \mathcal{F}^{(t)}] \geq M^{(t)}$ almost surely, we say $\{M^{(t)}\}$ is a submartingale.*

**Theorem 1** (Doob's Forward Convergence Theorem, (Williams, 1991)). *Let $\{M^{(t)}\}_{t \geq 1}$ be an $L^1$-bounded submartingale (in Def. 1). Then, almost surely, the limit $M^{(\infty)} = \lim_{t \to \infty} M^{(t)}$ exists and is finite.*

**Theorem 2** (Doob's Optional Stopping Theorem, (Williams, 1991)). *Let $\{M^{(t)}\}_{t \geq 1}$ be a submartingale (in Def. 1) and let $\tau$ be a stopping time. Then $\mathbb{E}[M^{(\tau)}] \geq \mathbb{E}[M^{(0)}]$ if any one of the following conditions hold: (i) $\tau$ is bounded, (ii) $P[\tau < \infty] = 1$ and $\{M^{(t)}\}$ is bounded for $\forall t \in \mathbb{Z}^+$, (iii) $\mathbb{E}[\tau] < \infty$ and $|M^{(t)} - M^{(t-1)}|$ is bounded for $\forall t \in \mathbb{Z}^+$.*

**Theorem 3** (Dominated Convergence Theorem, (Williams, 1991)). *Let $M^{(t)}$, $M$ be random variables s.t. $M^{(t)} \to M$ almost surely. There exists a random variable $Y \in L^1$ s.t. for $\forall t \in \mathbb{Z}^+$, $|M^{(t)}| < Y$ almost surely, then $\mathbb{E}[M^{(t)}] \to \mathbb{E}[M]$ as $t \to \infty$.*

**Lemma 1** (Minimum of Stopping Times, (Freeman & Stephenson, 2025)). *Let $\tau_1$ and $\tau_2$ be stopping times w.r.t. a filtration $\{\mathcal{F}^{(t)}\}$. Then $\tau_1 \wedge \tau_2 = \min(\tau_1, \tau_2)$ is also a $\{\mathcal{F}^{(t)}\}$ stopping time.*

# 4 OUR PROPOSED FRAMEWORK

In this study, we introduce a new framework, AdaInit, designed to mitigate BP issues in QNNs by leveraging generative AI (GenAI) models, particularly LLMs. Our **key innovations** can be described as follows. **(i)** First, unlike conventional one-shot initialization strategies, we propose a generative approach that iteratively generates effective initial model parameters $\boldsymbol{\theta}_0^* \in \mathbb{R}^{LNR}$ for QNNs that yield non-negligible gradient variance $\mathrm{Var}[\partial E]$, thereby mitigating BPs. In each iteration, we employ an LLM to refine the posterior (candidate initial model parameters $\boldsymbol{\theta}_0$) through adaptive prompting. After the posterior refinement, we train the QNN initialized with the generated $\boldsymbol{\theta}_0$ and further compute its $\mathrm{Var}[\partial E]$ in the early stage of training for monitoring BPs. The benefit of using LLM to refine the posterior is that the LLM can incorporate diverse textual instructions via prompts and adaptively update the prompts based on feedback from the previous iteration. This adaptive refinement allows our framework to dynamically optimize the generation process. **(ii)** To validate the generation quality, we employ Expected Improvement (EI), $\Delta^{(t)}$, as a guiding metric for the iterative process. Furthermore, we rigorously prove that the process satisfies the properties of a submartingale. Consequently, we theoretically establish the boundedness, thereby demonstrating that our proposed framework will ultimately find effective initial model parameters for QNNsin a finite step.

We present our framework workflow in Fig. 2 and further introduce details in Algo. 1. Given a GenAI model $f(\cdot)$, prompts $x_p$ for the $f(\cdot)$, a QNN $g(\cdot)$, and the number of iterations $T$, we first initialize $f(\cdot)$, $x_p$ (**line 1**) and also create an empty list $\varnothing$ for $\Theta_0^*$ to collect candidates of QNN's initial model parameters (**line 2**). After initialization, we run $T$ iterations for generation (**line 3**). In each iteration, let's say in the $t$-th iteration, we first employ $f(\cdot)$ with prompts $x_p^{(t)}$ and a prior distribution $P(\boldsymbol{\theta}_0^{(t)})$ to refine the posterior distribution $P(\boldsymbol{\theta}_0^{(t)}|x_p^{(t)})$, which is the generated initial model parameter $\boldsymbol{\theta}_0^{(t)}$ for the QNN (**line 4**). After generation, we train the QNN $g(\boldsymbol{\theta}_0^{(t)})$ with certain training epochs and compute the gradient variance $\mathrm{Var}[\partial E^{(t)}]$, whose gradient is abbreviated from $\frac{\partial E(\boldsymbol{\theta}^{(t)})}{\partial \boldsymbol{\theta}^{(t)}}$, where $\boldsymbol{\theta}^{(t)}$ denotes the QNN's model parameter in the $t$-th iteration (**line 5**). After computing the variance, we evaluate the improvement using the Expected Improvement (EI) metric, com-

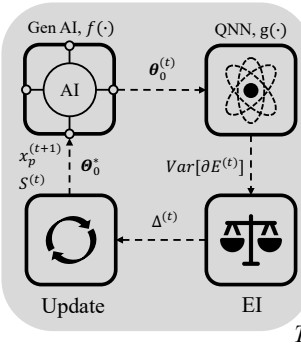

Figure 2: Our proposed framework follows an iterative process over $T$ iterations (gray area). In $t$-th iteration, we perform four sequential steps: (i) Generate $\boldsymbol{\theta}_0^{(t)}$ using a Gen AI model, $f(\cdot)$, (ii) Compute $\mathrm{Var}[\partial E^{(t)}]$ after QNN's training, (iii) Calculate EI, $\Delta^{(t)}$, and (iv) Update prompts $x_p^{(t+1)}$, historical maximum gradient variance $S^{(t)}$, and effective candidates $\boldsymbol{\theta}_0^*$ for next iteration. Dashed arrows indicate data flow and corresponding outputs in each step.

---

**Algorithm 1** Iteratively generate effective initial parameters for QNNs using a generative model.

---

**Require:** A GenAI model $f(\cdot)$, prompts $x_p$, a QNN $g(\cdot)$, the number of iterations $T$, a parameter $K$.
1: Initialize prompts $x_p$ and the GenAI model $f(\cdot)$;
2: Create an empty list $\Theta_0^* \leftarrow \varnothing$ to collect effective candidates of initial model parameters for the QNN, $g(\cdot)$;
3: **for** $t = 1$ to $T$ **do**
4:     $P(\boldsymbol{\theta}_0^{(t)}|x_p^{(t)}) \leftarrow f(x_p^{(t)}|\boldsymbol{\theta}_0^{(t)})P(\boldsymbol{\theta}_0^{(t)})$;
5:     $\mathrm{Var}[\partial E^{(t)}] \leftarrow g(\boldsymbol{\theta}_0^{(t)})$;
6:     $\Delta^{(t)} \leftarrow \max(\mathrm{Var}[\partial E^{(t)}] - S^{(t-1)}, 0)$;
7:     **if** $\Delta^{(t)} \geq \frac{1}{poly(N,L)K}$ **then**
8:         $x_p^{(t+1)} \xleftarrow{\boldsymbol{\theta}^{(t)}, \mathrm{Var}[\partial E^{(t)}], S^{(t-1)}} x_p^{(t)}$;
9:         $S^{(t)} \leftarrow \mathrm{Var}[\partial E^{(t)}]$;
10:        $\Theta_0^* \leftarrow \Theta_0^* \oplus [\boldsymbol{\theta}_0^{(t)}]$;
11:    **end if**
12: **end for**
13: **return** $\Theta_0^*$.

---

paring the current gradient variance $\mathrm{Var}[\partial E^{(t)}]$ to the historical maximum gradient variance, which is the cumulative sum of EI when EI meets the following conditions (**line 6**). If the current EI, $\Delta^{(t)}$, is effectively improved, i.e., $\Delta^{(t)} \geq 1/(poly(N,L)K)$, where $1/(poly(N,L)K)$ (with a parameter $K$ be determined later) denotes a strictly positive lower bound on the gradient variance of an $N$-qubit, $L$-layer QNN for each iteration, in the absence of BPs (**line 7**), then we update the prompts for the next iteration based on the current initial model parameters $\boldsymbol{\theta}_0^{(t)}$, the current gradient variance $\mathrm{Var}[\partial E^{(t)}]$, and the historical maximum gradient variance $S^{(t-1)}$ (**line 8**). After updating prompts,

we update the historical maximum $S^{(t)}$ for the next iteration, where (if $\Delta^{(t)} \geq 1/(poly(N,L)K)$) $S^{(t)} = S^{(t-1)} + \Delta^{(t)} = \text{Var}[\partial E^{(t)}]$ (**line 9**) and further concatenate $\boldsymbol{\theta}_0^{(t)}$ to the candidate list $\boldsymbol{\Theta}_0^*$ (**line 10**), which will be returned at the end (**line 13**). If so, the most effective initial model parameter $\boldsymbol{\theta}_0^*$ will be the last element in the candidate list. We can see that the input parameter $K$ is somewhat related to a wanted increment in $\text{Var}[\partial E^{(t)}]$, which is further linked to the desired underlying property provided by the GenAI model. This connection is explicitly shown in the theoretical analysis below.

**Analysis of time and space complexity.** Our framework runs $T$ iterations. In each iteration, posterior refinement, which is linearly related to the output size of $\boldsymbol{\theta}_0$, takes $\mathcal{O}(|\boldsymbol{\theta}_0|)$ for a fixed-size QNN. Besides, training $g(\boldsymbol{\theta}_0)$ with $T_{tr}$ epochs may take $\mathcal{O}(T_{tr} \cdot |\boldsymbol{\theta}_0|)$, where $T_{tr}$ denotes the number of training epochs for QNN. Combining $\boldsymbol{\theta}_0 \in \mathbb{R}^{LNR}$, the total **time complexity** is $\mathcal{O}(T \cdot (L \cdot N \cdot R + T_{tr} \cdot L \cdot N \cdot R)) \approx \mathcal{O}(T \cdot T_{tr} \cdot L \cdot N \cdot R)$. The space complexity primarily depends on the storage requirements. $\boldsymbol{\Theta}_0^*$ at most stores $T$ number of $\boldsymbol{\theta}_0$, which consumes $\mathcal{O}(T \cdot |\boldsymbol{\theta}_0|)$. The output of posterior refinement takes $\mathcal{O}(|\boldsymbol{\theta}_0|)$ space. Gradient variance and EI are scalars, which cost $\mathcal{O}(1)$ space. The prompts $x_p$ are iteratively updated and thus occupy $\mathcal{O}(|x_p|)$ space. Considering the size of $\boldsymbol{\theta}_0$, the total **space complexity** is $\mathcal{O}(T \cdot L \cdot N \cdot R + L \cdot N \cdot R + |x_p|) \approx \mathcal{O}(T \cdot L \cdot N \cdot R + |x_p|)$.

**Theoretical analysis of our framework.** We first present some necessary results and further discuss how those results can be interpreted. Full Proofs can be found in the **Appendix A**.

First, we define the Expected Improvement (EI) at each iteration $t$ as $\Delta^{(t)}$ and its cumulative sum in the past iterations as $S^{(t-1)}$ in Def. 2. Besides, we assume that the maximum possible gradient $\partial E_{max}$ during QNN's training is bounded by a positive constant $B_{\partial E}$, which is practical in real-world simulation. Next, we establish an upper bound for EI through Lem. 2 and Lem. 3.

**Definition 2** (Expected Improvement). *For $\forall\, t \in \mathbb{Z}^+$, the Expected Improvement (EI) in the $t$-th iteration is defined as:*

$$\Delta^{(t)} = \max(\text{Var}[\partial E^{(t)}] - S^{(t-1)}, 0),$$

*where $\text{Var}[\partial E^{(t)}]$ denotes the gradient variance in the $t$-th iteration, and $S^{(t-1)} = \sum_{t_i=1}^{t-1} \Delta^{(t_i)} \cdot I^{(t_i)}$ denotes the maximum observed gradient variance in the past iterations, where $I^{(t_i)}$ represents an indicator function $\mathbf{1}\big(\Delta^{(t_i)} \geq 1/(poly(N,L)K)\big)$ given a condition inside.*

**Assumption 1** (Bounded Maximum Gradient). *We assume there exists a positive constant $B_{\partial E} > 0$, s.t. the maximum possible gradient $\partial E_{max}$ during QNN's training satisfies:*

$$\big|\partial E_{max}\big| \leq B_{\partial E}.$$

*Without loss of generality, let's say $\partial E_{max} \in [-\frac{B_{\partial E}}{2}, \frac{B_{\partial E}}{2}]$.*

**Lemma 2** (Boundedness of Gradient Variance). *Given a certain-size quantum neural network (QNN), the variance of its gradient at the beginning of training, $\text{Var}[\partial E]$, is bounded by:*

$$\text{Var}[\partial E] \leq (\partial E_{\max} - \partial E_{\min})^2,$$

*where $\partial E_{\max}$ and $\partial E_{\min}$ denote the maximum and minimum values of the gradient $\partial E$, respectively.*

**Lemma 3** (Boundedness of EI). *From Def. 2 and Lem. 2, in the process of generating initial model parameters $\boldsymbol{\theta}_0$ for a certain-size QNN, for $\forall\, t \in \mathbb{Z}^+$, there exists a bound for the expected improvement (EI) s.t.*

$$\Delta^{(t)} \leq (\partial E_{max} - \partial E_{min})^2.$$

These results indicate that $S^{(t)}$ is $L^1$-bounded and integrable for each $t$. Building upon these lemmas, we investigate the submartingale property and rigorously prove in Lem. 4 that $S^{(t)}$ is a submartingale.

**Lemma 4** (Informal Statement of Submartingale Property). *Let $\{I^{(t)}\}_{t\geq 1}$ be a sequence of Bernoulli random variables on a probability space $(\Omega, \mathcal{F}, P)$ s.t. $I^{(t)} = 1$ with a real number $p \in (0, 1]$. Then, $\{S^{(t)}\}_{t\geq 1}$ is a submartingale with respect to the filtration $\{\mathcal{F}^{(t)}\}_{t\geq 1}$ which denotes the collections of all possible events up to time $t$.*

Note that the random variable is defined in relation to the comparison between the values of $\Delta^{(t)}$ and $1/(poly(N,L)K)$. From the definition, it is easy to see that $\Delta^{(t)} \geq 1/(poly(N,L)K)$ when $I^{(t)} = 1$, and $\Delta^{(t)} < 1/(poly(N,L)K)$ when $I^{(t)} = 0$. More precisely, each $\Delta \cdot I$ (omitting the superscripts) defines a joint distribution of one discrete random variable and one continuous random variable. More details about this intuition are provided in the proofs (**Appendix A**).

Leveraging the convergence of submartingales and the monotonicity of $S^{(t)}$, we establish in Lem. 5 that $S^{(t)}$ has a supremum, which indicates that our proposed framework can eventually generate effective initial model parameters for QNNs that can yield non-negligible gradient variance.

**Lemma 5** (Boundedness of Submartingale). *Let $\{S^{(t)}\}_{t \geq 1}$ be a submartingale w.r.t. a $\{\mathcal{F}^{(t)}\}_{t \geq 1}$ s.t. $\sup_t \mathbb{E}[|S^{(t)}|] < \infty$. Then, $\{S^{(t)}\}_{t \geq 1}$ is almost surely bounded by a finite constant $B_S$ s.t. $S^{(t)} \leq B_S, \quad a.s., \quad \forall t \in \mathbb{Z}^+$.*

Building upon Lem. 5, Thm. 4 shows the expected hitting time $\mathbb{E}[T_b]$ of a bounded submartingale, ensuring that our framework will converge to a desired solution within a finite number of iterations.

**Theorem 4** (Expected Hitting Time of a Bounded Submartingale). *Let $\delta = p/(poly(N,L)K) > 0$ with $p$ defined in Lem. 4. Let $T_b$ be the hitting time of a bounded submartingale $\{S^{(t)}\}_{t \geq 1}$, where $S^{(t)} \leq B_S$ almost surely, for $\forall t \in \mathbb{Z}^+$ (by Lem. 5). We define the hitting time as: $T_b = \inf \{t \in \mathbb{Z}^+ : S^{(t)} = b\}$, for some threshold $b \leq B_S$ such that the set is non-empty almost surely. Then the expected hitting time satisfies:*

$$\mathbb{E}[T_b] \leq \frac{b}{\delta} = \frac{b \cdot poly(N,L) \cdot K}{p}.$$

With this theorem, it is straightforward to derive the results for two meaningful cases: (i) $b = 1/poly(N,L)$; and (ii) $b = B_S$. This is summarized as Cor. 1 in the **Appendix A**. (i) When the threshold $b = 1/poly(N,L)$, the expected hitting time satisfies $\mathbb{E}[T_b] \leq K/p$, where the probability $p$ is defined in Lem. 4. Note that $K$ determines the desired threshold on increment $\Delta^{(t)}$. This is related to the generative power provided by the input GenAI model $f$, which is manifested in Lem. 4. Concretely, if the model $f$ can help return a wanted $\Delta^{(t)}$ with greater $1/(poly(N,L)K)$ (i.e. smaller $K$) and better probability (i.e. $p$), then the expected stopping iteration $T$ is getting smaller. (ii) When $b = B_S$, i.e., the supremum of the submartingale, $\mathbb{E}[T_b] \leq B_S \cdot poly(N,L) \cdot K/p$. This suggests that even when iteratively generating more effective initial parameters for QNNs (can yield higher gradient variance), our framework can still converge within a tractable (polynomial) number of iterations if $B_s = \mathcal{O}(poly(N,L))$. Both cases demonstrate that our framework is not only theoretically grounded but also robust in generating meaningful initializations for QNNs under different optimization goals.

Furthermore, we discuss extreme cases when $p$ is negligible, e.g., $p = \mathcal{O}(2^{-N})$. One such case involves ansatz-induced BPs (Holmes et al., 2022), where initialization-based methods fail in this scenario. Another case arises when LLMs repeatedly generate identical ineffective outputs. The former case could be addressed by modifying the QNN architecture (Holmes et al., 2022), while the latter one can be solved by adjusting hyperparameters, like temperature or top-p, to enhance output diversity (Achiam et al., 2023).

## 5 EXPERIMENT

In this section, we first introduce the experimental settings and further present our results in detail.

**Experimental settings.** We evaluate our proposed method across four public datasets, **Iris**, **Wine**, **Titanic**, and **MNIST**. We present the dataset statistics and settings in the **Appendix B**. In the experiment, we analyze the trend of gradient variance by varying the number of qubits, ranging from 2 to 20 in increments of 2 (fixed 2 layers), and the number of layers, spanning from 4 to 40 in steps of 4 (fixed 2 qubits). To obtain reliable results, we repeat the experiments five times and present them as curves (mean) with their bandwidth (standard deviation). Overall, our framework can generate effective model parameters within 50 iterations. In each iteration, we employ an Adam optimizer with a learning rate of 0.01 and a batch size of 20 to train a QNN with 30 epochs and compute the gradient variance. After training, we compute the expected improvement (EI) and compare it with

an assumed lower bound, $1/(poly(N,L)K)$, in each iteration, where we set $K = T$ in the experiments. We empirically choose the lower bound by $1/(T \cdot N^6)$ (**Appendix B**) and apply it for all cases, as we observe in Fig. 3 that the magnitudes of gradient variances are comparable across all datasets. For evaluation, we follow (McClean et al., 2018) to assess BP by measuring the gradient variance during the early stage of QNNs' training. A higher gradient variance implies a reduced likelihood of encountering BPs.

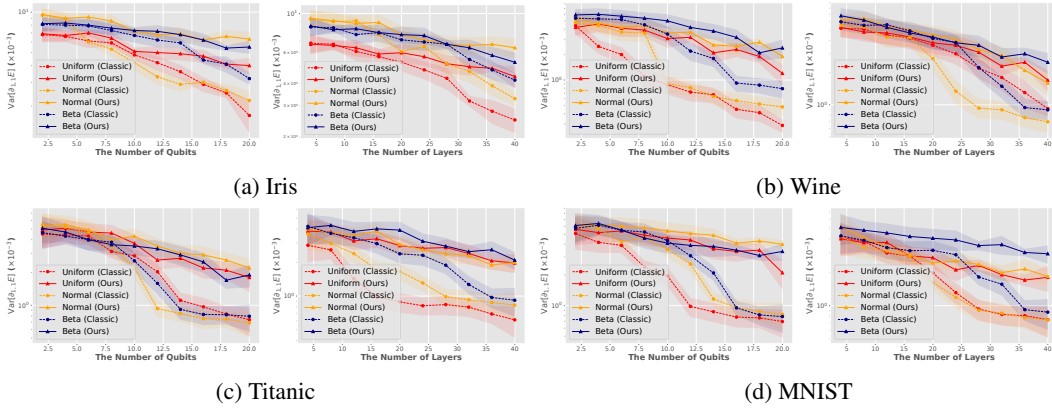

(a) Iris                (b) Wine

(c) Titanic              (d) MNIST

Figure 3: Analysis of gradient variance trends in the first element of QNNs' model parameters across varying qubit and layer settings for three classic initialization distributions, uniform, normal, and beta. "Classic" denotes that we initialize the model parameters with a classic distribution. "Ours" denotes that we use our framework to generate initial model parameters.

**Generating initial model parameters of QNNs using our framework can help mitigate BPs.** We analyze gradient variance trends in the first element of QNNs' model parameters across varying qubit and layer settings for three classic initialization distributions, uniform, normal, and beta distributions, which are presented in Fig. 4 as examples. For each initialization with classic distribution, we compare it ("Classic") with our proposed methods ("Ours"). As presented in Fig. 3, we observe that in the case of using classic initialization, the gradient variance of QNNs will significantly decrease as the number of qubits

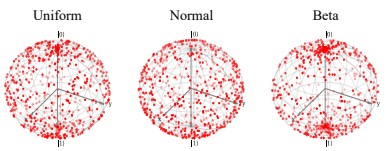

Figure 4: Example of three classic distributions commonly used for initialization. In the figure, the red dots represent the initial values of the model parameters.

or layers increases. Compared with it, our method can maintain higher variances, indicating that our framework can mitigate BPs better. In the rest of the experiments, if there is no specific state, we adopt a uniform distribution as prior knowledge for posterior refinement. Besides the above comparison, we further investigate the contribution of LLMs in the **Appendix B**.

**Comparison of generative performance using LLMs.** In our framework, the initial model parameters of QNNs are generated by LLMs. We compare the generative performance under varying QNN structures, such as different numbers of qubits or layers. Specifically, we primarily evaluate whether the correct size of model parameters can be generated by testing 20 combinations in accuracy, fixing either 2 layers while varying qubits from 2 to 20, or 2 qubits while varying layers from 4 to 40. As shown in Tab. 1, the results indicate that both GPT-4o and Claude 3.5 Sonnet can achieve 100% accuracy in generating the correct shapes of model parameters. Considering 4K output tokens are sufficient for our settings, we mainly use GPT-4o as the backbone LLMs. We provide additional observation in the **Appendix B**.

| LLMs | Acc. | Max i/o |
|---|---|---|
| GPT-4o | 100% | 128K / 4K |
| GPT-4o Mini | 85% | 128K / 16K |
| Gemini 1.5 Flash | 75% | 1M / 8K |
| Gemini 1.5 Pro | 90% | 2M / 8K |
| Claude 3.5 Sonnet | 100% | 200K / 8K |
| LLaMA 3 70B Instruct | 0% | 8K / 2K |
| LLaMA 3 405B Instruct | 50% | 128K / 2K |

Table 1: Comparison of initial parameters' generation by accuracy (Acc.) via GPT (Hurst et al., 2024), Gemini (Team et al., 2024), Claude (Anthropic, 2024), and LLaMA (Grattafiori et al., 2024). Suffix 'K' denotes 'thousand'.

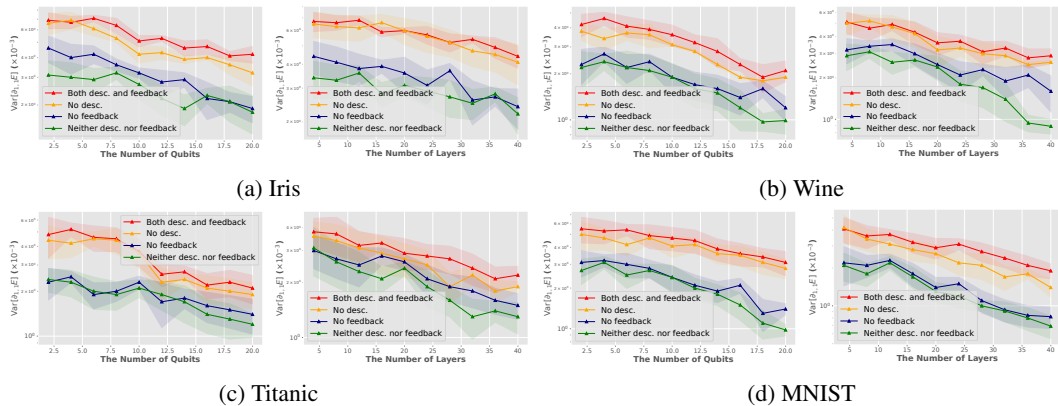

(a) Iris

(b) Wine

(c) Titanic

(d) MNIST

Figure 5: Analysis of prompts' impact, i.e., investigate whether data description (desc.) and gradient feedback (feedback) affect the gradient variance in the first element of QNNs' model parameters across different model structures, considering variations in the number of qubits and layers.

**Investigation of prompts.** We examine whether the content of prompts influences generative performance. In the experiments, we tested four prompting scenarios: (i) Including both data description and gradient feedback in prompts (Both desc. and feedback), (ii) Including gradient feedback only (No desc.), (iii) Including data description only (No feedback), (iv) Including neither data description nor gradient feedback (Neither desc. nor feedback). As the results presented in Fig. 5, we observe that suppressing either dataset description or gradient feedback in the prompts leads to a reduction in the gradient variance of QNNs. Notably, the reduction is more significant in most cases when gradient feedback is muted compared to the dataset description, suggesting that both factors play a crucial role in mitigating BPs, with gradient feedback contributing significantly more.

**Comparison with initialization-based strategies.**
We compare our framework with two representative initialization-based strategies, GaInit (Zhang et al., 2022) and BeInit (Kulshrestha & Safro, 2022). Both of them leverage well-designed Normal and Beta distributions to initialize the QNNs, respectively. For a fair comparison, we initialize the QNNs with the corresponding distribution. We present the results on Iris in Fig. 6 as an example. The results demonstrate that our framework can generate initial model parameters of QNNs that achieve higher gradient variance at the beginning of training as the model size increases, indicating better mitigation for BPs.

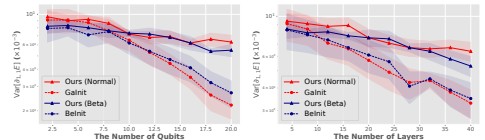

Figure 6: Comparison between two strategies and our framework, which is initialized with the corresponding data distribution for a fair comparison.

**Analysis of the expected improvement.** We analyze the patterns on the expected improvement (EI) and the corresponding gradient variance across various QNN structures as iterations progress. Representative experiments conducted on Iris are illustrated in Fig. 11 as an example (**Appendix B**). Our findings show that the framework can reliably discover meaningful initial parameters regardless of model size. Besides, as the model size grows, more iterations are required to obtain effective initial parameters that enable QNNs to maintain higher gradient variance. This is expected, as larger models expand the candidate space, demanding greater computational resources to explore effectively. Both observations verify the Cor. 1 (**Appendix A**).

**Sensitivity analysis of hyperparameters.** We analyze the sensitivity of hyperparameters, including Temperature and Top P, for LLMs. Temperature controls the randomness of predictions, with higher values generating more diverse outputs, while Top P affects the probabilities of token selections, balancing generation diversity and structural consistency. To identify optimal settings, we first narrowed down the ranges through manual tuning and then applied grid search to determine the best combinations (Temperature, Top P) for each dataset: Iris (0.5, 0.9), Wine (0.1, 0.45), Titanic (0.8, 0.75), and MNIST (0.8, 0.8), as presented in Fig. 7. The combinations of the above hyperparameters were used in this study.

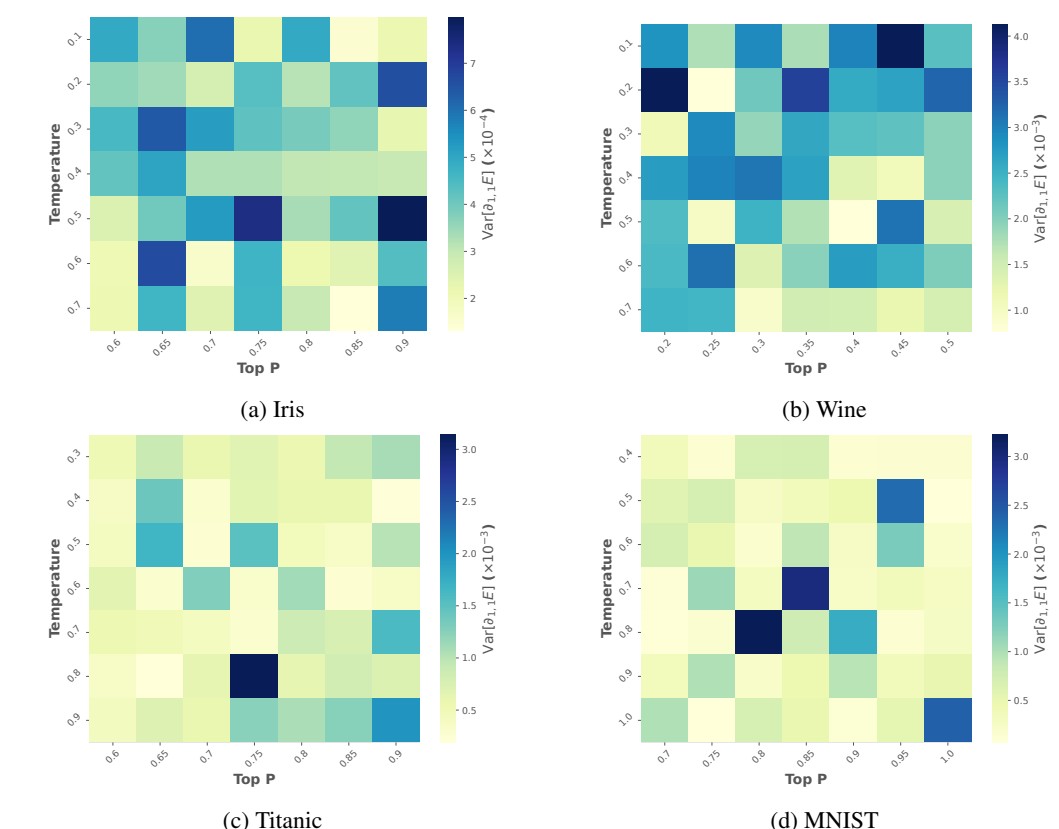

Figure 7: Analysis of the sensitivity of hyperparameters, including Temperature and Top P. The grid with the darkest color indicates the optimal combination.

Due to the page limit, we present our supplementary results, such as simulation time of QNNs, and the computational trade-off, in the **Appendix B**.

## 6 CONCLUSION

In this study, we aim to mitigate barren plateaus (BPs) by introducing a new AI-driven submartingale-based framework, namely AdaInit. This framework iteratively generates effective initial parameters using generative models, such as LLMs, for QNNs that yield non-negligible gradient variance, thereby mitigating BPs. Our theoretical analysis establishes the submartingale property for the iterative process, ensuring the effective generation. Through extensive experiments across various model scales, we demonstrated that AdaInit outperforms conventional classic initialization methods in maintaining higher gradient variance as QNN's sizes increase. Overall, this study might initiate a new avenue to explore how LLMs help mitigate BPs.

**Limitations, future work, and broad impact.** First, our theoretical analyses assume that the maximum gradient of QNNs is bounded by a positive constant, implying that gradient explosion does not occur during training, a condition that is typically satisfied in practice. Second, due to the practical limitations of quantum simulation, our experiments are constrained to QNNs with up to 20 qubits. We also assume an idealized setting where quantum measurements are noise-free. Moreover, our current scope excludes ansatz-induced BPs, which may be mitigated through architectural modifications, as discussed above. For **future work**, we plan to (i) accelerate the convergence of the iterative process and (ii) expand the applicability of our framework beyond BP mitigation. In particular, it can be leveraged to guide QNN architecture design or to identify optimal model parameters in training. More **broadly**, our framework can support robust QNN training across various domains, such as healthcare, where robustness and reliability are critical.

**Ethics Statement**

This work does not involve human subjects, sensitive personal data, or tasks with foreseeable negative societal impact. The datasets used (Iris, Wine, Titanic, and MNIST) are standard, publicly available benchmarks. We ensured compliance with their respective licenses and data usage guidelines. The proposed framework is designed to mitigate barren plateaus and does not directly enable harmful applications. Nonetheless, as with other advances in optimization, the method could potentially be applied in sensitive domains; in such cases, practitioners should carefully consider fairness, privacy, and security concerns in line with the ICLR Code of Ethics.

**Reproducibility Statement**

We have made every effort to ensure reproducibility of our results. Detailed theoretical proofs are included in Appendix A, whereas experimental settings, dataset splits, hyperparameters, architecture of the backbone quantum circuit, and the computing infrastructure are described in Appendix B. Besides, we provide sample code and datasets in the supplementary materials, which will be made publicly available upon publication. These resources, together with the descriptions of prompts, model parameters, and evaluation protocols, are intended to enable independent verification and extension of our work.

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

# Mitigating Barren Plateaus in Quantum Neural Networks via an AI-Driven Submartingale-Based Framework – Appendix

In the appendix, we present the proofs and the supplementary experiments in detail. **Datasets and sample code with a README file** are attached to the supplementary material for review purposes. These assets will be publicly available upon publication.

## A  PROOFS

In this section, we provide formal proofs that consolidate the theoretical guarantees of our framework.

*Proof for Lemma 2.* We denote a sequence of gradient $\partial \boldsymbol{E} = \{\partial E^{(t)}\}_{t=0}^{T_{tr}}$, where $T_{tr}$ represents the number of training epochs for a QNN. Within this sequence, we denote $\partial E_{max}$, $\partial E_{min}$, and $\overline{\partial E}$ as the maximum, minimum, and mean values of the gradient. For $\forall \, t \in \mathbb{Z}^+$, we have:

$$\partial E^{(t)}, \overline{\partial E} \in [\partial E_{min}, \partial E_{max}],$$

then the gap between $\partial E^{(t)}$ and $\overline{\partial E}$ will not exceed the range of $[\partial E_{min}, \partial E_{max}]$:

$$|\partial E^{(t)} - \overline{\partial E}| \leq \partial E_{max} - \partial E_{min}.$$

Thus, we have:

$$\begin{aligned}
\mathrm{Var}[\partial E] &= \frac{1}{T_{tr}} \sum_{t=1}^{T_{tr}} (\partial E^{(t)} - \overline{\partial E})^2 \\
&\leq \frac{1}{T_{tr}} \sum_{t=1}^{T_{tr}} (\partial E_{max} - \partial E_{min})^2 \\
&= (\partial E_{max} - \partial E_{min})^2.
\end{aligned}$$

Thus, the gradient variance $\mathrm{Var}[\partial E]$ satisfies the bound $\mathrm{Var}[\partial E] \leq (\partial E_{max} - \partial E_{min})^2$. $\qquad \square$

*Proof for Lemma 3.* From Def. 2, for $\forall \, t \in \mathbb{Z}^+$, in the $t$-th search iteration, we have:

$$\Delta^{(t)} = \max(\mathrm{Var}[\partial E^{(t)}] - S^{(t-1)}, 0).$$

Combining with Lem. 2, for $\forall \, t \in \mathbb{Z}^+$, we have:

$$\mathrm{Var}[\partial E^{(t)}], S^{(t-1)} \leq (\partial E_{max} - \partial E_{min})^2.$$

The above equation holds true as $S^{(t-1)}$ denotes the historical maximum gradient variance in the past iterations. Thus, we have:

$$\mathrm{Var}[\partial E^{(t)}] - S^{(t-1)} \leq (\partial E_{max} - \partial E_{min})^2,$$

which indicates that:

$$\Delta^{(t)} \leq (\partial E_{max} - \partial E_{min})^2.$$

$\qquad \square$

Before introducing the formal statement of submartingale property, we first define the random variables. Let $\alpha = 1/(poly(N,L)K)$. To avoid confusion, notably, we clarify that the $K$ refers to a previously defined parameter, whereas in Tab. 1, the suffix 'K' attached to numbers represents the unit 'thousand'. Formally, we define discrete random variables $I^{(t)}$ and continuous random variables $\Delta^{(t)}$ as follows:

$$\begin{cases} P(I^{(t)} = 1) = P(\Delta^{(t)} \geq \alpha) = p, \\ P(I^{(t)} = 0) = P(\Delta^{(t)} < \alpha) = 1 - p, \end{cases} \tag{5}$$

with a real number $p \in (0, 1]$. Hence, $I^{(t)}$ is a Bernoulli random variable as an indicator of the event $\Delta^{(t)} \geq \alpha$, and its associated continuous random variable $\Delta^{(t)}$ is implicitly defined with an arbitrary probability density function $p(y)$ that satisfies

$$P(\Delta^{(t)} \geq \alpha) = \int_\alpha^\infty p(y)dy = p. \tag{6}$$

With such a relationship between them, the following can be easily verified, for all $t \in \mathbb{Z}^+$,

$$P(I^{(t)} = x | \Delta^{(t)} = y) = \begin{cases} x, & y \geq \alpha \\ 1 - x, & \text{o.w.} \end{cases}. \tag{7}$$

**Lemma 6** (Submartingale Property, formal statement of Lemma 4). *Let $\{I^{(t)}, \Delta^{(t)}\}_{t \geq 1}$ be a sequence of random variables as defined in Eq. (5). We define joint random variables $W^{(t)} = \Delta^{(t)} \cdot I^{(t)}$ and the natural filtration $\mathcal{F}^{(t)} = \sigma(W^{(1)}, \cdots, W^{(t)})$. Then the sequence $\{S^{(t)}\}_{t \geq 1}$ as defined by Def. 2 is a submartingale with respect to the filtration $\{\mathcal{F}^{(t)}\}_{t \geq 1}$.*

*Proof for Lemma 6.* According to Def. 1, a process $S^{(t)}$ is a submartingale relative to $(\Omega, \mathcal{F}, P)$ if it satisfies Adaptedness, Integrability, and Submartingale.

**Adaptedness.** We first aim to verify that $S^{(t)}$ is determined based on the information available up to past $t$ iterations. By Def. 2, $S^{(t)} = \sum_{t_i=1}^t \Delta^{(t_i)} \cdot I^{(t_i)} = \sum_{t_i=1}^t W^{(t_i)}$ is a finite sum of random variables that are measurable w.r.t. $\sigma(W^{(1)}, \ldots, W^{(t)})$ (or $\sigma(I^{(1)}, \ldots, I^{(t)})$ for short due to the relationship between $I^{(t)}$ and $\Delta^{(t)}$ as shown in Eq. (5)). Thus, $S^{(t)}$ is also measurable w.r.t. $\mathcal{F}^{(t)}$, ensuring the adaptedness.

**Integrability.** In Lem. 3, $\Delta^{(t)} \leq (\partial E_{max} - \partial E_{min})^2$ for $\forall t \in \mathbb{Z}^+$. Thus,

$$\begin{aligned}
\mathbb{E}[|S^{(t)}|] &= \mathbb{E}\Big[\Big|\sum_{t_i=1}^t \Delta^{(t_i)} \cdot I^{(t_i)}\Big|\Big] \\
&\leq \mathbb{E}\Big[\Big|\sum_{t_i=1}^t (\partial E_{max} - \partial E_{min})^2 \cdot I^{(t_i)}\Big|\Big] \\
&< \infty,
\end{aligned}$$

which ensures $\mathbb{E}[|S^{(t)}|]$ is integrable for each $t$.

**Submartingale.** Before proving this condition, we show the following necessary inequality: with $\alpha = {}^1/_{(poly(N,L)K)}$,

$$\begin{aligned}
\mathbb{E}[\Delta^{(t)} \cdot I^{(t)}] &= \sum_{x=0,1} \int_{-\infty}^\infty P(I^{(t)} = x | \Delta^{(t)} = y) \cdot p(y) \cdot x \cdot y \, dy \\
&= \int_\alpha^\infty p(y) \cdot y \, dy \\
&\geq \alpha \int_\alpha^\infty p(y)dy \\
&= \alpha p, \tag{8}
\end{aligned}$$

where the second step follows from Eq. (7) and the last step uses Eq. (6). Apparently $\alpha p > 0$.

We observe that

$$S^{(t)} = S^{(t-1)} + \Delta^{(t)} \cdot I^{(t)}.$$

Since $S^{(t-1)}$ is $\mathcal{F}^{(t-1)}$-measurable, thus,

$$\begin{aligned}
\mathbb{E}[S^{(t)} | \mathcal{F}^{(t-1)}] &= \mathbb{E}[S^{(t-1)} + \Delta^{(t)} \cdot I^{(t)} | \mathcal{F}^{(t-1)}] \\
&= S^{(t-1)} + \mathbb{E}[\Delta^{(t)} \cdot I^{(t)}] \\
&\geq S^{(t-1)} + \alpha p \\
&\geq S^{(t-1)},
\end{aligned}$$

where the last two step applies Eq. (8).

Thus, the submartingale condition holds true for $\forall\, t \geq 1$ s.t.

$$\mathbb{E}\big[S^{(t)}\big|\mathcal{F}^{(t-1)}\big] \geq S^{(t-1)}, \quad \forall\, t \geq 1.$$

$\square$

Intuitively, $S^{(t)}$ tracks the cumulative amount of non-trivial variance improvements observed over the iterations — akin to measuring meaningful progress in exploration.

*Proof for Lemma 5.* Since the process $\{S^{(t)}\}_{t \geq 1}$ is a $L^1$-bounded submartingale s.t. $\sup_t \mathbb{E}[|S^{(t)}|] < \infty$, we apply the Doob's Forward Convergence Theorem (by Thm. 1), which guarantees the almost sure existence of a finite random variable $S^{(\infty)}$ s.t. $S^{(\infty)} = \lim_{t \to \infty} S^{(t)}$. This implies that the process $\{S^{(t)}\}$ has a well-defined almost sure limit.

Furthermore, if $\{S^{(t)}\}$ is monotone increasing, i.e., $S^{(t)} \leq S^{(t+1)}$, a.s., $\forall\, t \in \mathbb{Z}^+$, then the limit $S^{(\infty)}$ serves as a supremum for the entire process. By Defining $B_S := \sup_t S^{(t)} = S^{(\infty)}$, we obtain a desired bound $S^{(t)} \leq B_S$, a.s., $\forall\, t \in \mathbb{Z}^+$. $\square$

*Proof for Theorem 4.* To analyze the expected hitting time, we **first** construct a drift-adjusted process $\{Z^{(t)}\}_{t \geq 1}$ adapted to a filtration $\{\mathcal{F}^{(t)}\}_{t \geq 1}$ as $Z^{(t)} = S^{(t)} - \delta t$, where $\delta = p/(poly(N,L)K) > 0$.

Given $\alpha = 1/(poly(N,L)K)$ and follow those similar steps in the proof for Lem. 6, we can derive

$$\mathbb{E}[\Delta^{(t)} \cdot I^{(t)}] \geq \alpha p = \delta, \tag{9}$$

where the last step is by definition of $\delta$.

We then verify that $Z^{(t)}$ is also a submartingale:

• **Adaptedness**: Similr to $S^{(t)}$, $Z^{(t)}$ is also determined by the past $t$ iterations w.r.t. the same filtration $\sigma\big(W^{(1)}, \ldots, W^{(t)}\big)$ as $S^{(t)}$. Thus, $Z^{(t)}$ can meet the adaptedness.

• **Integrability**: Given that $S^{(t)}$ is $L^1$-bounded, and $Z^{(t)}$ is obtained by subtracting a deterministic finite value $\delta t$ from $S^{(t)}$, it follows immediately that $Z^{(t)}$ is also $L^1$-bounded, i.e., $\mathbb{E}[|Z^{(t)}|]$ is integrable for each $t$.

• **Submartingale**: We further show that $Z^{(t)}$ meets the submartingale inequality as follows.

$$\begin{aligned}
\mathbb{E}[Z^{(t+1)} \mid \mathcal{F}^{(t)}] &= \mathbb{E}[S^{(t+1)} - \delta(t+1) \mid \mathcal{F}^{(t)}] \\
&= \mathbb{E}[S^{(t+1)} \mid \mathcal{F}^{(t)}] - \delta(t+1) \\
&= \mathbb{E}[S^{(t)} + \Delta^{(t+1)} \cdot I^{(t+1)} \mid \mathcal{F}^{(t)}] - \delta(t+1) \\
&= S^{(t)} + \mathbb{E}[\Delta^{(t+1)} \cdot I^{(t+1)} \mid \mathcal{F}^{(t)}] - \delta(t+1) \\
&\geq S^{(t)} + \delta - \delta(t+1) \\
&= Z^{(t)},
\end{aligned}$$

where the fifth step is obtained by combining the fact that $\Delta^{(t+1)} \cdot I^{(t+1)}$ is independent of $\mathcal{F}^{(t)}$ and Eq. (9).

**Second**, we define the hitting time $T_b$ as $T_b = \inf\big\{T \in \mathbb{Z}^+ : S^{(T)} = b\big\}$. Without loss of generality, we assume $b \leq B_S$ since $S^{(t)} \leq B_S$ a.s., for $\forall\, t \in \mathbb{Z}^+$ (by Lem. 5). We further verify that $T_b$ is a bounded stopping time as follows.

We observe that $\{T_b \leq T\} = \{\exists\, t \leq T \text{ such that } S^{(t)} = b\} = \bigcup_{t=0}^{T}\{S^{(t)} = b\}$. Since $S^{(t)}$ is $\mathcal{F}^{(T)}$ measurable for $\forall\, t \leq T$, we have $\{S^{(t)} = b\} \in \mathcal{F}^{(T)}$, which indicates that the finite union $\bigcup_{t=0}^{T}\{S^{(t)} = b\} \in \mathcal{F}^{(T)}$. Hence, $\{T_b \leq T\} \in \mathcal{F}^{(T)}$, which by definition shows that $T_b$ is a bounded stopping time.

**Third**, we define $T_b \wedge t$ as $\min(T_b, t)$. Given that $T_b$ is a bounded stopping time, $T_b \wedge t$ is also a bounded stopping time (by Lem. 1). Based on this condition, the Doob's Optional Stopping Theorem implies that $\mathbb{E}[Z^{(T_b \wedge t)}] \geq \mathbb{E}[Z^{(0)}] = 0$ (by Thm. 2). Thus, we have:

$$\mathbb{E}[S^{(T_b \wedge t)} - \delta(T_b \wedge t)] \geq 0$$

implying

$$\mathbb{E}[S^{(T_b \wedge t)}] \geq \delta\mathbb{E}[T_b \wedge t].$$

Since $S^{(t)}$ is non-decreasing and bounded by $B_S$, we have $T_b \wedge t = T_b$ as $t \to \infty$ almost surely, which implies that $\mathbb{E}[T_b \wedge t] = \mathbb{E}[T_b]$. Moreover, by the definition of $T_b$, it follows that $S^{(T_b \wedge t)} \to b$ as $t \to \infty$, i.e., $\mathbb{E}[S^{(T_b \wedge t)}]$ is bounded by a dominating constant $b$. So, by the Dominated Convergence Theorem, as $t \to \infty$, we have $\mathbb{E}[S^{(T_b \wedge t)}] \to \mathbb{E}[S^{(T_b)}] = b$ (by Thm. 3).

By integrating the above equations and taking the limit, we conclude:

$$\mathbb{E}[T_b] \leq \frac{b}{\delta} = \frac{bK \cdot poly(N, L)}{p}.$$

$\square$

Thus, the following result can be derived immediately by plugging concrete values of $b$ into Thm. 4.

**Corollary 1** (Expected Hitting Time Under Specific Thresholds)**.**

1. *With an expected number of $K/p$ iterations, Algo. 1 can identify a candidate model parameter $\boldsymbol{\theta}_0^*$ that has $\mathrm{Var}[\partial E] \approx 1/poly(N,L)$.*

2. *With an expected number of $B_S \cdot K \cdot poly(N,L)/p$ iterations, Algo. 1 can identify a candidate model parameter $\boldsymbol{\theta}_0^*$ that has $\mathrm{Var}[\partial E] = \mathcal{O}(poly(N, L))$.*

# B SUPPLEMENTARY EXPERIMENTS

In this section, we present supplementary details about our experimental results.

**Dataset.** We evaluate our proposed method across four public datasets that are widely used in quantum machine learning. **Iris** [1] is a classic machine-learning benchmark that measures various attributes of three-species iris flowers. **Wine** [2] is a well-known dataset that includes 13 attributes of chemical composition in wines. **Titanic** [3] contains historical data about passengers aboard the Titanic and is typically used to predict survival. **MNIST** [4] is a widely used small benchmark in computer vision. This benchmark consists of $28 \times 28$ gray-scale images of hand-written digits from 0 to 9. We follow the settings of BeInit (Kulshrestha & Safro, 2022) and conduct

| Dataset | $|D|$ | $|F|$ | $|C|$ | Splits |
|---------|-------|-------|-------|--------|
| **Iris** | 150 | 4 | 3 | 60:20:20 |
| **Wine** | 178 | 13 | 3 | 80:20:30 |
| **Titanic** | 891 | 11 | 2 | 320:80:179 |
| **MNIST** | 60,000 | 784 | 10 | 320:80:400 |

Table 2: Statistics of datasets. $|D|$, $|F|$, and $|C|$ denote the original number of instances, features, and classes, respectively. "Split" denotes the split instances for the train, validation, and test data.

experiments in binary classification. Specifically, we randomly sub-sample a certain number of instances from the first two classes of each dataset to create a new subset. After sub-sampling, we employ the t-SNE technique to reduce the feature dimensions to ensure they do not exceed the number of available qubits. The statistics of the original datasets, along with the data splits for training, validation, and testing, are presented in Table 2. Importantly, the total number of sub-sampled instances corresponds to the sum of the split datasets. For instance, in the Iris dataset, the total number of sub-sampled instances is 100.

---

[1] https://archive.ics.uci.edu/ml/datasets/iris (Fisher, 1936)

[2] https://archive.ics.uci.edu/ml/datasets/iris (Fisher, 1936)

[3] https://www.kaggle.com/c/titanic (Kaggle, 2012)

[4] http://yann.lecun.com/exdb/mnist/ (LeCun et al., 2010)

**Contribution of LLMs.** Besides comparing our framework with the classic method, we further investigate LLMs' contribution to the initialization process on Iris. Specifically, we compare the generator within our framework when initialized using a random initializer (RI), which uniformly generates the parameters, versus using LLM-based uniform initialization (LLMs). As shown in Fig. 8, "RI" performs comparably to, or even worse than, the "Classic" method (uniform), as it generates random parameters without any guided refinement. In contrast, "LLMs" consistently outperforms both baselines, achieving significantly higher gradient variance. These findings suggest that the LLM-driven generator can more effectively explore the parameter space and identify better initializations within a limited number of iterations.

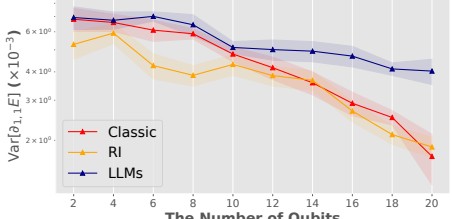

Figure 8: Comparison of model parameter initializations using a classic method, random initializer (RI), and LLMs. All methods apply a uniform distribution.

**Observations of open-source models.** We observe that two LLaMA 3 open-source LLMs (Grattafiori et al., 2024) perform significantly worse. To further understand these failures, we present representative cases in Tab. 3 and observe that both models fail to generate correct shapes of model parameters, indicating that these open-source models struggle to follow precise structural instructions and suggesting their current limitations in shape-constrained generation tasks.

| LLaMA 3 | Variables | Layer | Expected | Actual |
|---------|-----------|-------|----------|--------|
| 70B | $N \in [2, 20]$ | 0 | $(2, N, 3)$ | $(2, 3)$ |
| 70B | $L \in [4, 40]$ | 0 | $(L, 2, 3)$ | $(L, 3)$ |
| 405B | $L \in [4, 40]$ | 1 | $(2, 2)$ | $(2, 2, 2)$ |

Table 3: Comparison of generated model parameters (layer, expected shape, and actual shape) between two open-source LLMs—LLaMA 3 70B Instruct and LLaMA 3 405B Instruct—evaluated under various numbers of qubits ($N$) or layers ($L$).

**Simulation time in QNN training.** We assess the simulation time of QNN training under varying model sizes (number of qubits, $N \in [2, 20]$) and subsampled MNIST dataset sizes (number of instances, $|D| \in [800, 4000]$). We train QNNs for 30 epochs and present the **average runtime per epoch**. When varying $N$, we fix the number of layers $L$ at 2; when varying $|D|$, we fix both $N$ and $L$ as 2. As presented in Fig. 9,

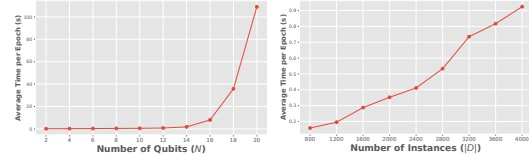

Figure 9: Assessment of the simulation time in QNN training.

in a classical simulated environment, the average training time of QNNs increases exponentially w.r.t. the number of qubits, while it grows roughly linearly with the dataset size. These observations reflect an inherent scalability issue in classical simulation of quantum systems. Such limitations are widely acknowledged in the quantum computing community and are unlikely to be fully overcome until practical quantum hardware becomes more accessible.

**Trade-off analysis.** To analyze the trade-off in Fig. 11, we present the relationship between computational cost (measured by the number of search iterations) and performance benefits (quantified by gradient variance) in Fig. 10. We observe a roughly linear relationship between them. In the 2-qubit case, 40% cost yields over 60% gain, while in the 20-qubit case, 35% cost yields over 44% gain, showing strong early-stage cost-effectiveness. The diminishing returns after a few iterations align with submartingale

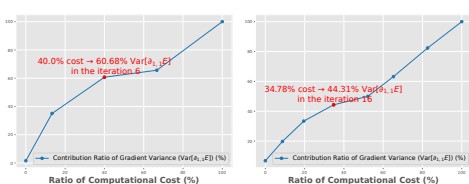

Figure 10: Trade-off analysis between computational cost and performance benefits for 2 qubits (left) and 20 qubits (right) setups.

optimization behavior, and the consistent trends across scales highlight AdaInit's suitability for budget-aware scenarios.

**Patterns of Expected Improvement (EI).** Due to limited pages, we present the patterns of EI in Fig. 11.

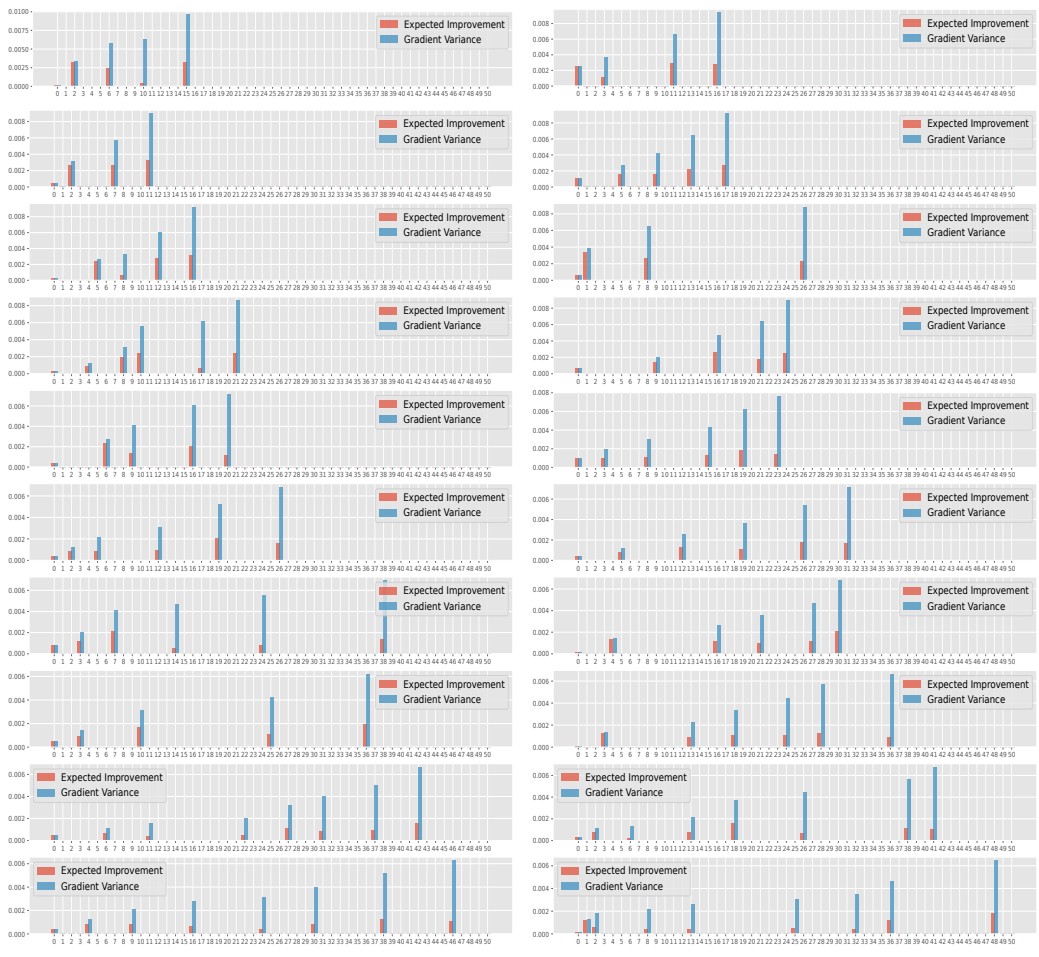

(a) The number of qubits ranges from 2 to 20.   (b) The number of layers ranges from 4 to 40.

Figure 11: We analyze the patterns of expected improvement and the corresponding gradient variance and present the results in two columns: the left column illustrates the trends w.r.t. the number of qubits, while the right column captures the effects of increasing the number of layers.

**Empirical analysis of the assumed lower bound.** To determine the assumed lower bound, $1/(poly(N,L)K)$, we conduct a trade-off analysis. A larger polynomial coefficient enlarges the admissible regime, but at the cost of including cases with vanishingly small gradient variance, whereas a smaller coefficient may filter out meaningful expected improvements, thereby preventing the framework from effectively exploring initial parameters. To proactively mitigate BPs, we restrict our attention to the range of qubits that are particularly susceptible to BPs. As illustrated in Fig. 12, we vary the polynomial coefficient and compare against the exponential baseline $1/(K2^{2N})$. Considering the trade-off, we empirically select $1/(KN^6)$ as the lower bound.

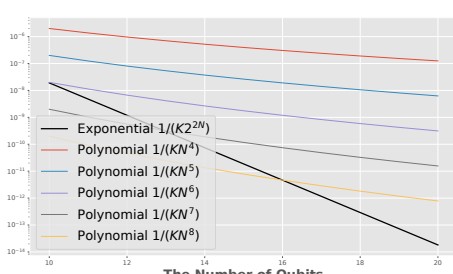

Figure 12: Trade-off analysis of the assumed lower bound, comparing polynomial terms against the exponential baseline, shown on a log scale.

**Prompt designs.** Before presenting the prompts, we first introduce the notation for the hyperparameter in the prompts. 'nlayers', 'nqubits', 'nrot', 'nclasses' denote the number of layers, qubits, rotation gates, and classes for the QNN, respectively. 'init' denotes the initial data distribution for the QNN. 'data_desc' denotes the data description. 'feedback' denotes the gradient feedback from the previous iteration.

**Prompts**

**Role:** data generator.
**Goal:** Generate a dictionary iteratively with the following shape:

```
{
  'l0': a list, shape=(nlayers, nqubits, nrot),
  'l1': a list, shape=(out_dim, nqubits),
  'l2': a list, shape=(out_dim)
}
```

**Requirements:**

- Data shape: nlayers={nlayers}, nqubits={nqubits}, nrot={nrot}, out_dim={nclasses}.

- Data type: float, rounded to four decimals.

- Data distribution: numerical numbers in each list are sampled from standard {init} distributions, which may be modeled from the following dataset.

- Dataset description: {data_desc}

- Adjust the sampling based on feedback from the previous searches: {feedback}

- Crucially, ensure that the length of 'l0' = 'nlayers' and the length of 'l1' = 'out_dim'.

- Print out a dictionary [only] (Don't show Python code OR include '["'python\n]', '["'json\n]', '["']').

**Model architecture of the quantum circuit.** In this study, we evaluate our framework using a backbone QNN consisting of a quantum circuit followed by a fully connected layer. Classical data are first encoded into quantum states via angle encoding, where each feature is mapped to rotation gates (e.g., $R_X$) on a specific qubit. This encoding maps data into the Hilbert space while preserving differentiability. The circuit applies repeated layers of parameterized rotations ($R_X$, $R_Y$, $R_Z$) and linear-topology CNOT gates for entanglement. After computation, the quantum state is measured in the computational

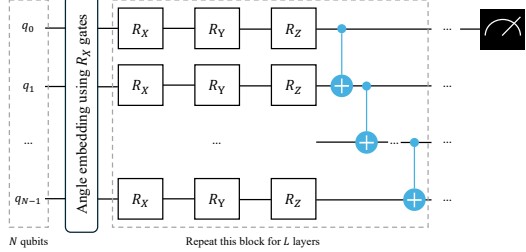

Figure 13: Architecture of our backbone quantum circuit. The number of rotation gates in this study is fixed as 3.

basis, and expectation values of Pauli-Z operators are computed and used as circuit outputs. These values are then processed by the classical fully connected layer. The overall architecture is adaptable in terms of layers, qubits, and rotation gates, as illustrated in Fig. 13.

**Hardware and software.** The experiment is conducted on a server with the following settings:

- Operating System: Ubuntu 22.04.3 LTS
- CPU: Intel Xeon w5-3433 @ 4.20 GHz
- GPU: NVIDIA RTX A6000 48GB
- Software: Python 3.11.8, PyTorch 2.2.2, Pennylane 0.35.1.

Based on the above computational infrastructure and setup, for example, our search framework can be reproduced in about 15 hours using 18 qubits.

**Use of LLMs.** LLMs were used only to assist in polishing the language and improving readability. No part of the technical content, analysis, or experimental results was generated by LLMs.

