# OpenReview forum: "Mitigating Barren Plateaus in Quantum Neural Networks via an AI-Driven Submartingale-Based Framework"
_ICLR.cc/2026/Conference — ICLR 2026 Conference Withdrawn Submission_

### Official Review · Reviewer_S2bm · 2025-10-22

**Soundness:** 1
**Presentation:** 1
**Contribution:** 1
**Rating:** 2
**Confidence:** 4

**Summary:**

The authors propose a framework based on LLMs to mitigate the BP phenomenon in QNNs. The authors provide a theoretical basis and first results to showcase their approach.

The paper has significant structural issues, in particular lacks a reading flow. The theorems and lemmas are stated as a long list without transitions and explanations, making the paper hard to follow. There is a substantial amount of literature on BPs missing and, considering that it is the main objective of the paper, needs to be added. Further, the related works are given as a list of papers, again, making it hard for a reader to actually grasp what the state-of-the-art is.

The experimental setup is not clearly described, and the experiments are not extensive enough to conclude about the performance of the method. In particular, starting initialization only at five distinct points in time does not result in any sort of statistical significance. It is for this reason that I do not think that the manuscript is in a publishable format.

Further, I do not think that this paper provides any contribution to the broader research community. Using LLMs to aid in optimizing a QNN, whose task the LLMs could solve by themselves without any problem, seems to be a fundamental overkill. There is a significant waste of computational resources while prompting state-of-the-art (!) LLMs to optimize a QNN to solve iris, wine, Titanic and MNIST. These datasets are further not reasonable datasets for validating the approach, since even the simplest regression can solve them up to high accuracy. I am more than aware that it is not possible yet to really solve classically hard ML problems with QML, but at least moving beyond trivially solvable datasets would be appropriate.

I would urge the authors to rethink the objective of this work or provide a convincing reason why employing LLMs would be a valuable approach to training QNNs.

**Strengths:**

- The authors test with a variety of different LLMs to evaluate differences in specific model used
- The authors put a lot of effort into providing the figures, which are as a result very intuitive to read

**Weaknesses:**

- "such as BPs, referring to a kind of..". This is really not a scientific description of a phenomenon, and in particular it is also not accurate towards the severity of the issues.
 - [Comment] please use \citep when citing within a sentence, this makes the paper more readable.
 - There is a substantial amount of literature on BPs missing. Given that the whole works aims at mitigating them, a clearer picture of state-of-the-art is necessary in the related work section. I would also expect more explanations of the approaches that are listed, s.t., the interested reader actually gets an overview of state-of-the-art, rather than just a collection of papers.
 - Line 110: "A quantum algorithm works by applying a sequence of unitary operators to an initial state". This is a description of a quantum circuit, not an algorithm. This is way too simplistic - it misses measurement, classical post-processing etc., which are crucial in quantum algorithms.
- Line 109: A state does not collapse into $\ket{i}$ for all $i$ after measurement, but rather into one of them.
- Line 112: VQCs do not play a core role in QNNs - QNNs are a subcategory of VQCs. If your definition is different, please elaborate.
- From Definition 1. Please provide some connection between the theorems and some elaboration on them. Making the paper a pure list of theorems makes it unreadable. The same holds for Definition 2 onwards. I would also suggest providing at least proof sketches in the main part of the paper.
- Line 306: ansatz-induced BPs are not a thing (ansatz-induced could also mean entanglement induced, etc.). The referenced paper talks about the connection of expressivity (or more concretely approximate 2-designs) to BPs.
- Missing quantitative results. How much better are the approaches in percent, how significant are the results?
- The authors state that they repeat the results five times to get reliable results, however, this is not backed by statistical evidence. To get anything close to reliable, one would (1) need to incorporate significance tests and (2) run a much larger number of random initializations.
- The authors provide a trade-off analysis in the Appendix, to compare the computational costs and performance benefits. I am missing the discussion on usage of LLMs here.

**Questions:**

- Line 118: Where is this definition of QNNs coming from? What kind of neural network layers are you talking about (repeated applications of U?). Where is the reference?
- Why does your architecture use a classical fully connected NN in the end? This is not standard to the best of my knowledge (otherwise provide reference). In any case, this should be stated in the main part of the paper.
- In light of the choice of adding a FCNN at the end of the quantum circuit, I would be curious to see how the performance of the models is if only the FCNN is used for prediction with the classical data only as input. As I said, I am questioning the whole setup, but even having QNN + classical FCNN to predict the simple datasets used seems already an overkill before employing the LLMs.

---

> ### Author Response · Authors · 2025-11-25
>
> **W1&W2: Writing & Format**\
> We appreciate your suggestion regarding the writing and citation format. We have revised the manuscript to use \citep throughout the text to improve readability.
>
> **W3: Coverage of Related Work**\
> Due to page limits, we prioritized works most relevant to our approach. However, we are happy to expand the discussion and include any specific references you suggest in the revision.
>
> **W4: Description of Quantum Algorithms (Line 110)**\
> Thanks for pointing out this potential confusion. This paragraph is intended to introduce basic concepts, while VQCs are detailed separately in a subsequent section. We have removed this sentence to avoid confusion.
>
> **W5: Description of the State Collapse  (Line 109)**\
> We agree that the phrase "for all i" may bring confusion. We have removed this phrasing.
>
> **W6&Q1: Definition of VQCs vs. QNNs (Line 112)**\
> In this work, we distinguish between VQCs (referring to quantum circuits whose model architecture is constructed solely from parameterized quantum circuits without interleaving classical neural network layers) and QNNs (hybrid models combining quantum circuits with classical layers). The definition of QNNs as hybrid models (quantum circuits combined with classical layers) is a standard paradigm in the NISQ era. This architecture is foundational to concepts like Quantum Circuit Learning (Mitarai et al., 2018) and Transfer Learning in hybrid networks (Mari et al., 2020). We have added a clarification in Line 111 to explicitly state this definition and further added references in line 112 to support the definition of QNNs.
>
> **W7: Presentation of Definitions and Theorems**\
> We first introduce the necessary statistical tools in the "Tools from Probability Theory" section (Line 141). Then, in the "Theoretical Analysis of our Framework" section (Line 239), we explicitly describe the logical connections between Definition 2 and the subsequent Lemmas to ensure a coherent narrative rather than a mere list of theorems.
>
> **W8: Clarification on Terminology and Scope**\
> We used the term "Ansatz-induced BPs" to refer to the phenomenon described by Holmes et al. (2022), where the expressibility of the ansatz is the root cause of vanishing gradients. This type of BP is structural in nature, so that initialization-based strategies are ineffective, and architectural modifications are required. In this work, we don’t cover the "Ansatz-induced BPs”. To avoid confusion, we modified the term to "expressibility-induced BPs”.
>
> **W9: About Quantitative Results**\
> We follow the settings by McClean et al. (2018), using gradient variance (GVar) as the primary metric. A higher GVar directly indicates a more trainable landscape and the mitigation of Barren Plateaus. Our results (e.g., Fig. 3) quantitatively show that our approach maintains GVar orders of magnitude higher than classical initialization methods as the model scales increase.
>
> **W10: About Statistical Evidence**\
> Reporting mean±std is standard practice in the ML community to indicate stability. Since our focus is on validating effectiveness, this format sufficiently demonstrates robustness without the need for additional significance tests.
>
> **W11: About Trade-off Analysis and LLMs**\
> The trade-off analysis in the Appendix primarily focuses on the relationship between computational cost (measured by search iterations) and performance benefits (quantified by gradient variance). Regarding the specific impact and usage of LLMs in this process, we have provided detailed analyses in Lines 365, 396, 426, 864, and 879.
>
> **Q2: Why Use a Classical Fully-connected Layer at the End?**\
> We adopted the model setting from Zhuang et al. (2024) (cited in our related work). The primary objective of this study is to verify whether our framework effectively mitigates Barren Plateaus in the quantum component. The inclusion of fully-connected layers is part of the standard hybrid architecture and does not alter the fundamental gradient behavior of the quantum circuit or our main conclusions regarding initialization strategies.
>
> **Q3: Use FCNN as the Backbone Model**\
> Our work focuses on mitigating BPs for QNNs rather than optimizing the model in the classification tasks. While a classical neural network might currently achieve higher accuracy on these simple datasets, this does not diminish the motivation to study and improve the loss landscape of quantum models for future scalability.
>
> **References**
> - Mitarai, K., Negoro, M., Kitagawa, M., & Fujii, K. (2018). Quantum circuit learning. Physical Review A, 98(3), 032309.
> - Mari, A., Bromley, T. R., Izaac, J., Schuld, M., & Killoran, N. (2020). Transfer learning in hybrid classical-quantum neural networks. Quantum, 4, 340.
> - Zhuang, Jun, Jack Cunningham, and Chaowen Guan. "Improving trainability of variational quantum circuits via regularization strategies." arXiv preprint arXiv:2405.01606 (2024).

---

> > ### Comment · Reviewer_S2bm · 2025-11-25
> >
> > I would like to thank the authors for the clarification.

---

> > > ### Author Response · Authors · 2025-11-26
> > >
> > > Thanks for your prompt response. We're happy to clarify if you have further questions.

---

### Official Review · Reviewer_PM6V · 2025-10-29

**Soundness:** 2
**Presentation:** 2
**Contribution:** 2
**Rating:** 2
**Confidence:** 4

**Summary:**

This paper proposes AdaInit, an AI-driven framework that mitigates barren plateaus in quantum neural networks. The method iteratively generates effective initialization parameters using large language models (LLMs) guided by a submartingale-based process to ensure non-vanishing gradient variance. The authors provide theoretical guarantees for convergence and boundedness, and conduct experiments on various benchmark datasets, including Iris, Wine, Titanic, and MNIST.

**Strengths:**

The paper introduces a novel intersection of LLMs and quantum optimization, supported by rigorous theoretical proofs, but I have not checked them. The submartingale formulation is mathematically grounded, and experiments are clearly organized.

**Weaknesses:**

1. The use of a general-purpose LLM for updating parameter probabilities appears unjustified. It is unclear why an LLM not trained on quantum circuit data can reliably identify parameters that avoid barren plateaus.
2. The authors claimed the time complexity for QNN training is O(T_{tr} \cdot |\theta_0|), while overlooking the cost of calculating gradients in each step. In particular, evaluating gradients accurately may require exponentially many measurements, making the method potentially infeasible at larger scales.
3. The proposed algorithm also relies on computing the gradient variance $Var(\partial E^{(t)})$ at each iteration to guide parameter updates. Similarly, this calculation again demands exponentially many measurements under barren plateau conditions, raising concerns about its scalability and practicality on current NISQ hardware.
4. The manuscript includes numerous lemmas and theorems, some of which are peripheral to the main idea. This dense theoretical presentation obscures the core contributions. The paper would benefit from presenting only key theoretical results in the main text and moving technical proofs and auxiliary lemmas to the appendix for clarity.
5. The experimental setup is somewhat limited for studying barren plateaus, as the authors only explore configurations with either many qubits and shallow depth or few qubits and deep depth. Barren plateaus are theoretically less likely to occur in these regimes. To convincingly demonstrate the framework’s effectiveness, the authors should evaluate it on larger quantum circuits with both high qubit counts and significant depth, where barren plateaus are more probable and challenging to mitigate.

**Questions:**

The questions are included in the Weakness.

---

> ### Author Response · Authors · 2025-11-25
>
> We sincerely thank the reviewer PM6V for the insightful comments, particularly regarding the applicability of LLMs, the scalability of the simulation, and the theoretical presentation. We address your concerns point-by-point below.
>
> **W1: Rationale for Using General-Purpose LLMs without Training**\
> We clarify that our framework does not require the LLM to possess pre-trained knowledge of quantum physics. As discussed in Sec. 4 (line 177), the core advantage of using LLMs lies in their ability to **refine the posterior distribution** through in-context learning. Unlike static initialization methods, LLMs can incorporate diverse textual instructions via prompts and adaptively update these prompts based on scalar feedback (gradient variance) from the previous iteration. Crucially, this process operates entirely at the inference stage; it does not require training or fine-tuning the model on quantum circuit data. The LLM acts as a reasoning engine to navigate the parameter space effectively based on history, rather than a domain-specific predictor.
>
> **W2: Limitations of Classical Simulation Scalability**\
> We acknowledge the reviewer's concern regarding the time complexity. As noted in our discussion on **Simulation time in QNN training** (line 890), the exponential increase in training time with respect to the number of qubits is an inherent limitation of simulating quantum systems on classical hardware. This bottleneck is widely acknowledged within the quantum computing community and is unlikely to be fully overcome until practical quantum hardware becomes accessible. Our complexity analysis was intended to reflect the algorithmic operations of the framework itself, while the simulation cost is an unavoidable environmental constraint.
>
> **W3: Computational Cost of Gradient Variance Monitoring**\
> Regarding the cost of computing gradient variance, we specifically calculate the variance during the **first 30 training epochs** of the initialization phase. We wish to clarify that the computational overhead for this specific monitoring step (calculating the variance of the collected gradients) does not increase exponentially with the number of qubits. It remains a manageable cost designed to provide the necessary "signal" for the submartingale process to detect an escape from the barren plateau.
>
> **W4: Theoretical Rigor as a Core Contribution**\
> We appreciate the suggestion to improve readability. However, we respectfully emphasize that the theoretical proofs are a core contribution of this work. **Lemma 4** (Submartingale Property) and **Theorem 4** (Expected Hitting Time) provide the mathematical guarantee that our iterative process will almost surely converge to a parameter region with non-negligible gradients within finite steps. This rigorous foundation distinguishes our AI-driven framework from pure heuristic or random search methods. Therefore, we believe it is essential to retain the core construction of the submartingale and the stopping time analysis in the main text to highlight the theoretical depth and soundness of our approach.
>
> **W5: Justification of Experimental Configurations**\
> Regarding the experimental settings, we relied on the formal definition of Barren Plateaus (Eq. 4, line 133), which establishes that the phenomenon is primarily driven by the increase in the **number of qubits (N)**. While we included variations in layer depth (L) for completeness, our experimental design intentionally **decouples these variables**. This allows us to clearly isolate and demonstrate the exponential degradation of gradient variance as the system scale (N) grows, which is the hallmark of the Barren Plateau problem. We believe this setup effectively validates the mitigation capabilities of our framework.

---

### Official Review · Reviewer_3PRx · 2025-10-29

**Soundness:** 2
**Presentation:** 3
**Contribution:** 2
**Rating:** 2
**Confidence:** 3

**Summary:**

The paper introduces AdaInit, designed to address the barren plateaus problem in QNN training. Unlike traditional methods that use static, preset distributions to initialize parameters, Adainit adopted an iterative strategy: it leverages a generative AI model to dynamically generate new initial parameters that are more likely to escape the BP problem. The paper theoretically models this iterative process as a sub martingale and proves its convergence. Experiments show that this method outperforms existing methods in maintaining high gradient variance, especially as the QNN scales.

**Strengths:**

1. Casts initialization as an iterative search with a simple progress statistic and connects it to sub martingale tools.
2. Empirical sweep spans qubits and layers across four datasets with ablations on prompts and llm hyperparams.

**Weaknesses:**

1.The core mechanism's assumptions are too strong. The paper's core argument rests on the assumption that large language models can intelligently explore the high-dimensional parameter space of QNNs, thereby finding regions of high gradient variance. This is an extremely strong assumption. The framework provides only a scalar feedback signal (gradient variance) from an extremely complex space. However, to date, there is no theoretical or empirical evidence demonstrating how LLMs can perform such high-dimensional, geometry-aware optimization.
2. Inconsistency between the theoretical framework and the proposed method: While the application of submartingale theory is ingenious, its underlying assumptions appear to contradict the very nature of the AdaInit framework. The generation of the new parameter θ(t) explicitly depends on historical data (θ(t-1) and S(t-1)), violating the independence assumption crucial to many theoretical assertions.
3. Critical omission of experimental noise: This paper aims to address the back propagation problem, a particularly acute problem in the NISQ era. However, the experiments in this paper are conducted in an idealized, noise-free simulation environment. This is a critical omission, as noise can create its own "barren plateaus" and fundamentally alter the gradient distribution. Methods that work in a noise-free environment are not guaranteed to work on real quantum hardware. The lack of experiments under realistic noise models severely limits the practical significance and impact of our results.

**Questions:**

1.Could you provide a more rigorous justification for why an LLM is a suitable tool for this task? What inductive biases or capabilities of modern LLMs make them superior to classical optimization algorithms like Bayesian Optimization or Evolutionary Strategies for exploring the QNN parameter space based on scalar feedback?
2.How do you reconcile the apparent contradiction between the i.i.d. assumption used in your submartingale proofs and the history-dependent, iterative nature of the AdaInit algorithm? Could you clarify how the theoretical guarantees hold under the more realistic condition of temporal dependency in the parameter generation process?

---

> ### Author Response · Authors · 2025-11-25
>
> We sincerely thank Reviewer 3PRx for the constructive feedback and provide our response below.
>
> **W1&Q1 About LLMs’ Assumption and Comparison**\
> First, prior research (Du et al. 2023) indicates that iterative inference in LLMs can converge to a fixed point (often referred to as 'Consensus'). Second, our framework leverages LLMs to refine generation based on feedback from the previous iteration, such as model parameters and gradient variance. Crucially, the LLM does not need to exhaustively explore the entire parameter space. Instead, its key advantage lies in its ability to learn from past failures and rectify subsequent generations, thereby identifying superior initial parameters for QNNs. We provide empirical results in Line 864 demonstrating that the LLM-based generator significantly outperforms a random generator. Furthermore, we provide a theoretical analysis within this work formally proving the convergence of our framework.
>
> Comparing with Bayesian Optimization (BO), whose primary goal is to construct a probabilistic surrogate model to handle complex objective functions (such as QNNs). In contrast, our framework does not aim to approximate the target model itself but focuses specifically on generating effective initial parameters for it. Furthermore, unlike Evolutionary Strategies (ES), which rely on fixed evolutionary strategies, our LLM-driven approach utilizes feedback to 'adaptively' synthesize and refine the distribution of initial parameters.
>
> **W2&Q2 About i.i.d. Assumption**\
> Thank you for pointing out the contradiction. We agree that i.i.d. is a strong assumption and is not necessary used in the martingale process. Removing such an assumption will not affect our proof and experimental results as our framework doesn’t depend on it. Therefore, we have removed “i.i.d.” in lines 267 & 713, and removed “$\{\Delta^{(t)}\}_{t \ge 1}$ is i.i.d.” in line 750. The manuscript has been updated.
>
> **W3 About “back propagation problem”**\
> Regarding “This paper aims to address the back propagation problem …”, in this work, we aim to address QNNs’ BP issues through a framework that leverages generative models, like LLMs, to synthesize candidate initial parameters for QNNs. Addressing the back propagation problem is out of scope in this work. Also, we assume an idealized setting where quantum measurements are noise-free (line 477).
>
> **Reference**\
> Du, Y., Li, S., Torralba, A., Tenenbaum, J. B., & Mordatch, I. (2023). Improving factuality and reasoning in language models through multiagent debate. In Forty-first International Conference on Machine Learning.

---

> ### Comment · Reviewer_3PRx · 2025-11-26
>
> Thanks for your responses. However, I still could not identify the benefits of leveraging LLM for QNN, which to me is more like a combination of two hot topics or keywords, rather than presenting a great number of potential follow-up works in both communities or research directions. So I decide to keep my score.

---

> > ### Author Response · Authors · 2025-11-27
> >
> > Thank you for your confirmation, and Happy Thanksgiving.
> > In terms of technical innovation, we want to clarify that our proposed framework is the first work that incorporates the submartingale property into the iterative inference of LLMs. Furthermore, we theoretically demonstrate that our framework, characterized by this submartingale property, exhibits favorable convergence. Consequently, this allows us to identify non-negligible gradient variance for Quantum Neural Networks (QNNs), effectively mitigating Barren Plateaus (BPs). Rather than merely offering incremental improvements on existing methods, our work explores QNN initialization from a new perspective.

---

### Official Review · Reviewer_L7YZ · 2025-11-01

**Soundness:** 3
**Presentation:** 2
**Contribution:** 1
**Rating:** 2
**Confidence:** 4

**Summary:**

The authors propose a foundational framework called AdaInit, which leverages generative models with a submartingale property to iteratively synthesize initial parameters for quantum neural networks (QNNs) that maintain non-negligible gradient variance, thereby effectively mitigating the Barren Plateau problem.

**Strengths:**

1.Utilizes generative models for parameter initialization, effectively ensuring non-negligible gradient variance in the early stage of training, thereby mitigating Barren Plateau (BP) issues.
2.Extensive experiments validate the framework’s effectiveness across different QNNs scales, demonstrating good scalability and adaptability.
3.The framework is adaptive, dynamically adjusting parameter generation based on dataset characteristics and gradient feedback, enhancing the intelligence of initialization.
4.Introduces a theoretically analyzable submartingale-based iterative process, providing formal guarantees for the convergence of generated parameters.

**Weaknesses:**

1.Limited experimental scale: Currently validated only on small-scale QNNs with few qubits.Suggestion: Future work could explore larger-scale QNNs or real quantum hardware to evaluate scalability in high-dimensional quantum systems.
2.Focuses only on initialization: In practical applications, the key metrics are final training performance, including accuracy, convergence speed, and robustness/generalization. AdaInit only addresses the ability to start training by avoiding Barren Plateaus. Suggestion: To address this limitation, future work could extend AdaInit to not only ensure non-negligible initial gradient variance but also consider downstream training performance
3.Restricted dataset types: Only binary classification tasks were used, limiting generality.Suggestion: Test on multi-class or higher-dimensional datasets to verify robustness and applicability.
4.Dependency on generative model quality: The effectiveness of initialization relies on the output of the LLM or generative model.Suggestion: Introduce validation mechanisms or enhance the feedback loop to ensure generated parameters are effective.
5.High computational cost of iterative generation: Multiple iterations for parameter generation and gradient variance evaluation can be time-consuming.

**Questions:**

1.The experiments are limited to small-scale QNNs (≤20 qubits) and binary classification datasets. How would AdaInit perform on larger QNNs or multi-class/high-dimensional datasets?
2. AdaInit mainly ensures non-negligible initial gradient variance, but does it improve final training performance (accuracy, convergence speed, robustness/generalization)?Is higher initial gradient variance always correlated with better downstream training outcomes? Could the model still converge slowly or achieve suboptimal final accuracy?
3. How would AdaInit perform in noisy intermediate-scale quantum (NISQ) devices with measurement errors?
4.AdaInit relies on generative models (e.g., LLMs) to produce parameters. How sensitive is the method to different LLMs, model sizes, or prompt designs?Could errors or instability in the generated parameters negatively affect QNN training?

---

> ### Author Response · Authors · 2025-11-25
>
> **W1&Q1: About Experimental Scale**\
> As discussed in the Simulation time in QNN training (line 890), we acknowledge the inherent constraints of simulating quantum systems on classical hardware. The training time for QNNs in a classical simulation environment increases exponentially with the number of qubits, a limitation that is widely acknowledged in the quantum computing community and unlikely to be fully overcome until practical quantum hardware becomes accessible. Currently, simulating QNNs with up to 20 qubits is computationally feasible, whereas exceeding this threshold renders classical simulation intractable.
>
> **W2&Q2: Focus on Initialization**\
> Our primary objective is to address the Barren Plateaus (BPs), determining whether QNNs are trapped in BPs at the initialization stage, rendering them untrainable. We follow the settings in McClean et al. (2018) to monitor the gradient variance. A higher gradient variance at the beginning of training implies a reduced likelihood of encountering BPs and a higher probability of trainability. It is important to note that achieving higher classification accuracy or faster convergence rates is outside the scope of this study, which focuses strictly on mitigation via initialization.
>
> **W3: Use of Binary Classification Tasks**\
> To investigate BP mitigation, our focus lies on whether QNNs of varying qubits (N) and depths (L) can maintain non-negligible gradients. Consequently, the complexity of the classification task itself is secondary. We employ binary classification on standard datasets as a proxy to evaluate the gradient landscape, as is common practice in BP literature.
>
> **W4&Q4: Dependence on Generative Model Quality**\
> Exploring the utilization of generative models, such as LLMs, to mitigate BPs is one of the core contributions of this work. To the best of our knowledge, this is the first work that leverages LLMs with a submartingale-based framework to address BP issues in QNNs.
> - We compared the generative capabilities of mainstream LLMs (line 365), confirming that GPT-4o achieves 100% generation accuracy with efficient resource usage.
> - Our framework incorporates a validation mechanism and feedback loop (Algo. 1, line 190) that iteratively refines the prompt based on prior results to generate parameters for QNN initialization.
> - We analyzed the sensitivity of different prompt designs, demonstrating the importance of including gradient feedback (Fig.5, line 378).
> - We also discussed failure cases (line 879), explicitly noting that models like LLaMA 3 are currently unsuitable for this specific generation task due to their inability to follow structural constraints.
>
> **W5: Computational Cost of Iterative Generation**\
> We provide a detailed time and space complexity analysis in Sec. 4 (line 223). This analysis indicates that the total time complexity is primarily determined by the scale of the QNN and the number of iterations. QNN training within the loop is solely for validating whether the initialization has escaped the BPs. The time cost of the framework itself is largely dictated by the API rate limits of the LLMs. It is crucial to distinguish that the scalability issues of QNN simulation are an inherent property of classical simulation, not a flaw in the proposed framework.
>
> **Q3: About Noise Measurements**\
> As stated in Limitations (line 477), we assume an idealized setting where quantum measurements are noise-free. The analysis of measurement noise falls outside the scope of this study.

---

### Author Response · Authors · 2025-12-03
**Note to Area Chair: Summary of Contributions and Responses**

We sincerely thank reviewers for their constructive feedback.

All reviewers acknowledge that our work targets a critical challenge in quantum neural networks (QNNs), namely **barren plateaus** (BPs), where the gradient variance will degrade exponentially as the number of qubits increases. To address this challenge, we **propose a new AI-driven submartingale-based framework**, AdaInit, that iteratively generates QNNs’ initial parameters using generative models such as LLMs while providing rigorous martingale-theoretic guarantees on boundedness and finite expected hitting time of non-negligible gradient variance. To the best of our knowledge, this is the **first work** that mitigates BPs in QNNs by leveraging generative models (e.g., LLMs) to drive an adaptive submartingale-based process for QNNs’ initialization with provable guarantees on maintaining non-negligible gradient variance within finite expected time.

We empirically **validate our framework** on four public datasets across a wide range of qubit and layer counts, where it consistently maintains higher gradient variance (our main evaluation metric) than classical static initializations (e.g., uniform, normal, and beta) and two representative BP mitigation methods, GaInit and BeInit. We also conduct ablations on prompt design, LLM choice, and sampling hyperparameters, further supporting our design choices.

The **main concerns** raised in reviews focus on the quantum scale in simulations. Such a limitation is widely acknowledged in the quantum computing community and unlikely to be fully overcome until practical quantum hardware becomes accessible.
In our response, we **address all concerns from reviewers** and **revise** our papers based on feedback.

Taken together, we believe the reviews support that the work is original, theoretically solid, and practically promising under realistic resource constraints. Therefore, we respectfully suggest that it be viewed as a **weak-accept candidate** that can stimulate further research on AI-assisted training of QNNs for mitigating BPs.

---

### Note · Authors · 2026-01-06

I have read and agree with the venue's withdrawal policy on behalf of myself and my co-authors.